



# Suspended Particulate Matter drives the spatial segregation of nitrogen turnover along the hyper-turbid Ems estuary

Gesa Schulz[1,2], Tina Sanders[1], Justus E. E. van Beusekom[2,3], Yoana G. Voynova[2], Andreas Schöl[4], Kirstin Dähnke[2]

[1]Institute of Geology, Center for Earth System Research and Sustainability (CEN), University Hamburg, Hamburg, 20146, Germany
[2]Institute of Carbon Cycles, Helmholtz-Zentrum Hereon, Geesthacht, 21502, Germany
[3]Institute of Oceanography, University Hamburg, Hamburg, 20146, Germany
[4]Department of Microbial Ecology, Federal Institute of Hydrology, Koblenz, 56068, Germany

*Correspondence to*: Gesa Schulz (Gesa.Schulz@hereon.de)

## Abstract

Estuaries are nutrient filters and change riverine nutrient loads before they reach coastal oceans. They have been extensively changed by anthropogenic activities like draining, deepening, and dredging to meet economic and social demand, causing significant regime changes like tidal amplifications and in some cases to hyper-turbid conditions. Furthermore, increased nutrient loads, especially nitrogen, mainly by agriculture cause coastal eutrophication. Estuaries can either act as a sink or as a source of nitrate, depending on environmental and geomorphological conditions. These factors vary along an estuary, and change nitrogen turnover in the system. Here, we investigate the factors controlling nitrogen turnover in the hyper-turbid Ems estuary (Northern Germany) that has been strongly impacted by human activities. During two research cruises in August 2014 and June 2020, we measured water column properties, dissolved inorganic nitrogen, dual stable isotopes of nitrate and dissolved nitrous oxide concentration along the estuary. Overall, the Ems estuary acts as a nitrate sink in both years. However, three distinct biogeochemical zones exist along the estuary. A strong fractionation (~ 26 ‰) of nitrate stable isotopes points towards nitrate removal via water column denitrification in the hyper-turbid Tidal River, driven by anoxic conditions in deeper water layers. In the Middle Reaches of the estuary nitrification gains in importance turning this section into a net nitrate source. The Outer Reaches are dominated by mixing with nitrate uptake in 2020.

We find that the overarching control on biogeochemical nitrogen cycling, zonation and nitrous oxide production in the Ems estuary is exerted by suspended particulate matter concentrations and the linked oxygen deficits.

## 1 Introduction

Estuaries can significantly alter riverine nutrient loads before they reach adjacent coastal oceans (Bouwman et al., 2013; Crossland et al., 2005). Estuaries have been extensively altered by humans and anthropogenic activities to meet economic and social demands. Draining, damming, diking, deepening and dredging lead to significant regime changes including tidal amplification, hyper-turbid conditions and loss of habitats (e.g. Stronge et al. 2005; Winterwerp et al. 2013; De Jonge et al.





2014). High nutrient loads from agriculture, waste water and urban runoff have induced eutrophication (Galloway et al., 2003;
Howarth, 2008; Van Beusekom et al., 2019), one of the greatest threats to coastal ecosystems worldwide (e.g. Howarth and
Marino 2006; Voss et al. 2011).
Depending on predominant microbial processes, environmental conditions and geomorphological characteristics, estuaries can
either act as a sink or as an additional source of nitrate (Dähnke et al., 2008; Middelburg and Nieuwenhuize, 2001). Especially
the balance between remineralisation/nitrification and denitrification determines the net role of a specific estuary. Previous
studies found that biogeochemical changes of  dissolved oxygen saturation, residence time or light penetration affect this
balance of nutrient uptake and removal (Carstensen et al., 2014; Diaz and Rosenberg, 2008; Thornton et al., 2007; Voss et al.,

41   2011).

To disentangle the role of nitrate production and removal processes, stable isotopes are a frequently used tool, because nitrogen
turnover processes usually discriminate versus heavier isotopes, leading to an enrichment in the pool of remaining substrate.
The magnitude of enrichment, the so-called isotope effect, is process-specific (e.g. Granger et al. 2004; Deutsch et al. 2006;
Sigman et al. 2009).
Nitrification and denitrification also produce nitrous oxide ($N_2O$) (Knowles, 1982; Tiedje, 1988; Wrage et al., 2001; Francis
et al., 2007), a potent greenhouse gas that contributes to global warming (IPCC, 2007). Estuaries are potential sources for
nitrous oxide (Bange, 2006) and,  together with coastal wetlands, contribute approximately 0.17 to 0.95 Tg $N_2O$-N per year to
the global nitrous oxide budget of 16.9 Tg $N_2O$-N per year (Murray et al., 2015; Tian et al., 2020). Numerous factors control
estuarine nitrous oxide emissions. Oxygen depletion, nutrient levels and possibly organic matter composition trigger nitrous
oxide production. Therefore, nitrous oxide emissions is linked to eutrophication (e.g. de Wilde and de Bie 2000; Galloway et
al. 2003; Murray et al. 2015; Quick et al. 2019). The role of nitrous oxide production can vary along an estuary, depending on
the environmental and geomorphological properties.
Although the individual nitrogen turnover processes are well understood, the interplay of multiple stressors on the nitrogen
cycle needs further investigation (e.g. Billen et al. 2011; Giblin et al. 2013; Sanders and Laanbroek 2018). Therefore, we
investigate how water column properties can change the nitrogen turnover, emerging eutrophication and nitrous oxide
production along an estuary.
We performed two research cruises along the Ems estuary, a heavily managed estuary in Germany that underlies anthropogenic
pressures from fertilizer input, dredging, and channel deepening (De Jonge, 1983; Talke and de Swart, 2006; Johannsen et al.,
2008) leading to a significant increase of suspended particulate matter concentration in the inner estuary (De Jonge et al.,
2014). We studied water column nutrient and stable isotope composition, as well as suspended particulate matter properties in
the Ems estuary to investigate spatial dynamics in nitrogen removal, nitrogen turnover processes and their relation to nitrous
oxide production. We have (1) evaluated the zonation of nitrogen turnover along the estuary, (2) identified the dominating
nitrogen turnover pathways in individual zones, and (3) discussed the controlling factors of nitrogen cycling and emerging
nitrous oxide production. Ultimately, with this study we provide a better insight into the effects of water column properties
and biogeochemistry on estuarine nutrient turnover.



## 2 Methods

### 2.1 Study site

The Ems estuary is situated on the Dutch-German border (Fig.1). The estuary is approximately 100 km long and stretches from
the weir at Herbrum to the island Borkum. The Ems discharges into the Wadden Sea, a part of the southern North Sea (Van
Beusekom and de Jonge, 1994). The catchment of the Ems is 17 934 km² (Krebs and Weilbeer, 2008) and is densely populated,
with 86 % urban/agricultural land-use (Johannsen et al., 2008). The Ems is also an important waterway with ports in Delfzijl
and Emden, and is used for transport of large vessels from the shipyard in Papenburg to the North Sea (Talke and de Swart,
74 2006).

The Ems is characterized by steep gradients in salinity and tides (Compton et al., 2017). It has an average discharge of
80.8 m³ s⁻¹, with low fresh water discharge in summer, and highest discharge between January and April. The Ems is a hyper-
turbid estuary with high suspended sediment concentrations (De Jonge et al., 2014; Van Maren et al., 2015b), reaching values
of up to 30-40 g L⁻¹ and more in fluid mud layers (Winterwerp et al., 2013). Channel deepening has led to tidal amplification
and an increased upstream sediment transport in the tidal Ems (De Jonge et al., 2014). The increase of suspended matter
lowered light penetration, and led to decreasing oxygen concentration (Bos et al., 2012). Bos et al. (2012) classified the Ems
estuary as a degraded ecological system with high nutrient loads.
Based on geomorphological characteristics, the Ems can be divided into four sections: the Tidal River (km 14–km 35), Dollard
Reach (km 35–km 43), Middle Reaches (km 43–km 75) and Outer Reaches (downstream from km 75) (Fig. 1).





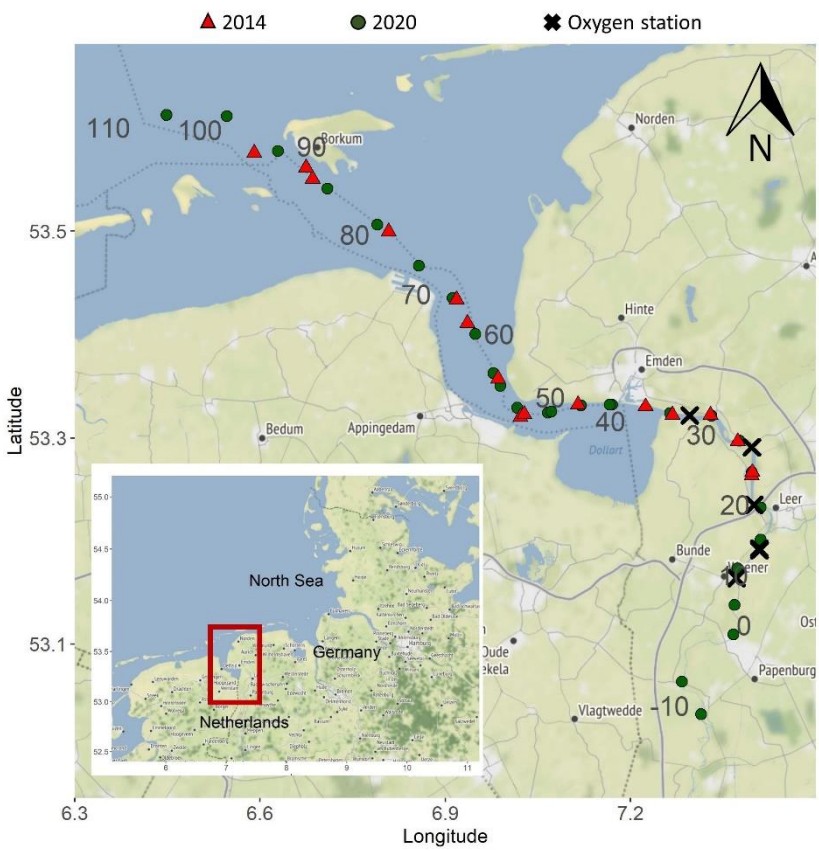

**Figure 1: Map of the Ems estuary displaying the sampling stations. Red triangles mark stations done in 2014, green points stations in 2020 and crosses show oxygen measurements stations. The grey numbers show the stream kilometers calculated according to German federal waterways (wsv.de). Background map: © OpenStreetMap contributors 2021. Distributed under the Open Data Commons Open Database License (ODbL) v1.0.**

## 2.2 Sampling

Water samples were taken during two research cruises with the research vessel *Ludwig Prandtl* in August 2014 and June 2020. Nutrient concentration and suspended particulate matter concentration from the cruise in 2014 have been published in Sanders and Laanbroek (2018). An onboard membrane pump provided the on-line in-situ FerryBox system with water from 2 m below the surface. The FerryBox system continuously measures dissolved oxygen, water temperature, pH, salinity, fluorescence and turbidity (Petersen et al., 2011). In 2014, the dissolved oxygen measurements from the FerryBox were about 32 µmol L$^{-1}$ lower than the Winkler titrations of two discrete samples collected in July 2014. This offset was used to correct the FerryBox optode measurements. Salinity measurements were checked using Optimare Precision Salinometer (Bremerhaven, Germany), and the error of the FerryBox measurements was within 0.01 salinity units.

Discrete water samples were taken from a bypass of the FerryBox system. The samples for nutrient and isotope analysis were filtered immediately through combusted, pre-weighted GF/F Filters (4 h, 450 °C), and stored frozen in acid-washed (10 %





HCl, overnight) PE-Bottles at -20 °C until analyses. The filters were stored at -20 °C for later analysis of suspended particulate
matter (SPM) (Röttgers et al., 2014), $\delta^{15}$N-SPM and C/N ratios. C/N ratios were measured with an Elemental Analyzer
(Eurovector EA 3000) calibrated against a certified acetanilide standard (IVA Analysentechnik, Germany). The standard
deviation was 0.05 % and 0.005 % for carbon and nitrogen respectively.  During the 2020 cruise, nitrous oxide gas phase mole
fractions were continuously measured in unfiltered water.

## 2.2 Dissolved oxygen measurements

During the cruises in 2014 and 2020, we measured dissolved oxygen concentration in surface water using the FerryBox system
(see above). For a more detailed view on oxygen dynamics, we also used data provided by the German Federal Institute of
Hydrology (Bundesanstalt für Gewässerkunde – BfG) (BfG, unpulished). Vertical profiles of oxygen concentration were taken
at four monitoring stations along the Ems estuary (Fig. 1) in August 2014 and June 2020 using an YSI 6660 probe. At these
stations, oxygen and temperature were also continuously measured with miniDot® (PME, Precision Measurement
Engineering) loggers at 0.5 m above the bottom at Ems kilometer 11.8 and 24.5 in 2014 and additionally at 18.2 and 33.0 in

112  2020.

## 2.3 Nutrient measurements

Nutrient concentration (nitrate, nitrite, ammonium, silicate and phosphate) was measured with a continuous flow auto analyzer
(AA3, SEAL Analytics) using standard colorimetric and fluorometric techniques (Hansen and Koroleff, 2007). Measurement
ranges were 0-400 µmol-N L$^{-1}$ for combined nitrate and nitrite, 0-17.8 µmol-N L$^{-1}$ for nitrite, 0.07-25 µmol-N L$^{-1}$ for
ammonium, 0-1000 µmol-Si L$^{-1}$ for silicate and 0-16.1 µmol-P L$^{-1}$ for phosphate.

## 2.4 Isotopic analysis

The stable isotope composition of nitrate ($\delta^{15}$N-NO$_3^-$, $\delta^{18}$O-NO$_3^-$) was measured using the denitrifier method (Sigman et al.,
2001; Casciotti et al., 2002), which is based on the isotopic analysis of nitrous oxide . In brief, *Pseudomonas aureofaciens*
(ATCC#13985) reduce nitrate and nitrite$^-$ in the filtered water samples to nitrous oxide. Nitrous oxide was measured by a
GasBench II coupled with an isotope ratio mass spectrometer (Delta Plus XP, Thermo Fisher Scientific). Two international
standards (USGS34, $\delta^{15}$N-NO$_3^-$ -1.8 ‰, $\delta^{18}$O-NO$_3^-$ -27.9 ‰; IAEA, $\delta^{15}$N-NO$_3^-$ +4.7 ‰, $\delta^{18}$O-NO$_3^-$ +25.6 ‰) and one internal
standard ($\delta^{15}$N-NO$_3^-$ +7.6 ‰, $\delta^{18}$O-NO$_3^-$ +24.4 ‰) were used to calibrate the samples. The standard deviation for standards
and samples was <0.2 ‰ (n= 4) and <0.5 ‰ (n=4) for $\delta^{15}$N-NO$_3^-$ and $\delta^{18}$O-NO$_3^-$ respectively. Nitrite concentration of the
samples was usually <5 %. When nitrite exceeded 5 %, it was removed prior to analysis using  Sulfamic Acid (Granger and
Sigman, 2009).
An Elemental analyzer (Carlo Erba NA 2500) coupled with an isotope ratio mass spectrometer (Finnigan MAT 252) was used
to measure $\delta^{15}$N-SPM values. IAEA N1 ($\delta^{15}$N = +0.4 ‰), IAEA N2 ($\delta^{15}$N = +20.3 ‰) and a certified sediment standard (IVA
Analysetechnik, Germany) were used as reference materials.



## 2.5 Equilibrator based nitrous oxide measurements and calculations

An nitrous oxide analyzer (Model 914-0022, Los Gatos Res. Inc.) coupled with a sea water/gas equilibrator measured the dry mole fraction of nitrous oxide and water vapor in the water column using off-axis integrated cavity output spectroscopy. The set-up and instrument precision is described in detail in Brase et al. (2017). The equilibration time of nitrous oxide of approximately 7 min was taken into account for data processing.

For validation of the measurements, we measured two standard gas mixtures of nitrous oxide in synthetic air regularly (500.5 ppb ± 5 % and 321.2 ppb ± 3 %). No drift was detected. For further data processing, we calculated 1 min averages of nitrous oxide detected dry mole fraction (ppm). We calculated the dissolved nitrous oxide concentration in water ($N_2O_{cw}$) using the Bunsen solubility function of Weiss and Price (1980) taking temperature differences between sample inlet and equilibrator into account (Rhee et al., 2009). Nitrous oxide saturation (s) was calculated using Eq. (1), based on nitrous oxide concertation in water ($N_2O_{cw}$) and atmospheric nitrous oxide ($N_2O_{air}$).

$$s = 100 \times \frac{N_2O_{cw}}{N_2O_{air}} \tag{1}$$

Atmospheric nitrous oxide was measured regularly during our cruise and was on average 0.33 ppm during our cruise in 2020. The gas transfer coefficient ($k$) was calculated based on Borges et al. (2004), where $u_{10}$ is wind speed 10 m above surface, and $Sc$ is the Schmidt number (Eq. (2)). Sea-to-air flux densities were calculated using Eq. (3).

$$k = 0.24 \times (4.045 + 2.58u_{10}) \times \left(\frac{Sc}{600}\right)^{-0.5} \tag{2}$$

$$f = k \times (N_2O_{cw} - N_2O_{air}) \tag{3}$$

## 2.5 Nitrate mixing calculations

Nitrate concentration from conservative mixing ($C_{Mix}$) between two endmembers was calculated for each sample using the classical mixing model of Liss (1976).

$$C_{Mix} = f \times C_R + (1 - f)C_M \tag{4}$$

Where $C_R$ and $C_M$ stand for the concentration of the riverine and marine end-members, respectively, and $f$ denotes freshwater fractionation in each sample calculated as follows:

$$f = \frac{(S_M - S_{Mix})}{(S_M - S_R)} \tag{5}$$

$S_{Mix}$, $S_M$, $S_R$ denote the salinity of the sample, marine and riverine endmembers, respectively. We used the concentration-weighted mean of the isotopic values of the marine ($\delta_M$) and riverine ($\delta_R$) end-members to calculate the theoretical isotope value of samples following conservative mixing ($\delta_{Mix}$) (Fry, 2002):



$$\delta_{Mix} = \frac{f \times C_R \times \delta_R + (1-f) \times C_M \times \delta_M}{C_{Mix}}$$ (6)
**2.6 Isotope effect**
During turnover processes, nitrogen isotopes ratios change along a specific isotope effect that helps to identify individual
process pathways (e.g. Kendall et al. 2007). Isotope effects were calculated with an open-system approach where the reactant
nitrate is continuously supplied and partially consumed, and steady state is assumed. This leads to a linear relationship between
isotope values of nitrogen and fraction $f$, where $f = ([C]/[C_{initial}])$. The isotope effect $\varepsilon$ corresponds to the slope of the regression
line (Sigman et al., 2009),
$$\varepsilon_{substrate} = \frac{\delta^{15}N_{substrate} - \delta^{15}N_{initial}}{(1-f)}$$ (7)
$$\varepsilon_{product} = \frac{\delta^{15}N_{product} - \delta^{15}N_{initial}}{f}$$ (8)
Where $\delta^{15}N_{substrate}$, $\delta^{15}N_{product}$, $\delta^{15}N_{initial}$ denote $\delta^{15}$N values of the substrate and product at the time of sampling and the initial
value. The remaining fraction of substrate at the time of sampling is described by $f$. In the present study, the mixing line
determines initial concentrations and isotope values.
**2.7 Statistical analysis**
All statistical analysis were done using R packages. Pearson correlation matrices were calculated with ggcorr from the
R-package GGally v.2.0.0 (GGally: Extension to "ggplot2," 2021). From the R-package stats v4.0.2 (The R Stats Package,
Version 4.0.2, 2021), we used the function prcomp for the principal component analysis (PCA). Salinity was not taken into
account for the multivariate analysis.
**3 Results**
**3.1 Hydrographic properties and dissolved nutrients in surface water**
To evaluate controls on nutrient cycling, we first regard the hydrochemical properties that were measured in 2014 and 2020 in
surface water, alongside with nutrient concentrations and nitrogen stable isotope composition (Fig. 2).





**Figure 2:** *Near surface water column properties along the Ems estuary: (a) ammonium concentration in (μmol L⁻¹), (b) nitrite concentration in (μmol L⁻¹), c) nitrate concentration in (μmol L⁻¹), (d) δ15N-Nitrate in (‰), (e) δ18O-Nitrate in (‰), (f) nitrous oxide saturation in (%), (g) salinity, (h) dissolved oxygen concentration in (μmol L⁻¹), (i) suspended particulate matter concentration (SPM) in (mg L⁻¹), (j) C/N ratios, (k) particulate organic carbon fraction (POC) in (%), (l) δ15N- suspended particulate matter in (‰). Red triangles mark stations done in 2014 and green points stations in 2020.*








Discharge ranged from 59.7 m³ s⁻¹ to 67.5 m³ s⁻¹ in 2014 and was ~ 30 m³ s⁻¹ in 2020. The long-term average discharge is
30-40 m³ s⁻¹ in June and August (NLWKN Bst. Aurich and Engels, 2021). The mean water temperature was 23 °C in 2014
and 17 °C in 2020. Salinity ranged from ~ 0.5 to ~ 32 in both years. In 2014, the sampling section started with the onset of the
salinity gradient (km 20), whereas the most upstream sample in 2020 was taken near Herbrum (km -14) (Fig. 2g), this and the
sample at stream kilometer -9 were taken by a bucket from the shore. The vessel based transect started in Papenburg (km 0).
Nitrate was the major form of dissolved inorganic nitrogen (DIN) and decreased with increasing salinity. Nitrate concentration
decreased from 177 µmol L⁻¹ to 3.9 µmol L⁻¹ in 2014 and from 166 µmol L⁻¹ to 4.9 µmol L⁻¹ in 2020 (Fig. 2c).
Ammonium (Fig. 2a) and nitrite (Fig. 2 b) concentration were generally low in the tidal river, with average concentrations of
~3 and 1 µmol L⁻¹, respectively. One sample (June 2020, stream kilometer 25) had an unusually high ammonium concentration
of 13 µmol L⁻¹. In the Dollard Reach, ammonium and nitrite concentration increased with salinity in 2020, whereas this
increase occurred further downstream, i.e., in the Middle reaches, in 2014. The highest ammonium concentration was similar
in 2014 and 2020, with 8.5 µmol L⁻¹ and 10.2 µmol L⁻¹ respectively. Whereas in 2020 nitrite concentration reached 1 µmol L⁻¹,
with little variability along the transect, in 2014, it reached a maximum of 3.5 µmol L⁻¹ in a distinct peak in the Middle and
Outer Reaches.
Incoming nitrate isotope values were elevated in the most upstream regions of the Tidal River with values of 15 ‰ for
$\delta^{15}N\text{-}NO_3^-$ and 6 ‰ for $\delta^{18}O\text{-}NO_3^-$ in 2020, and 17 ‰ and 8 ‰ for $\delta^{15}N\text{-}NO_3^-$ and $\delta^{18}O\text{-}NO_3^-$ in 2014. Isotope values increased
further to a local maximum of 25 ‰ and 11 ‰ for $\delta^{15}N\text{-}NO_3^-$ and $\delta^{18}O\text{-}NO_3^-$ around km 13 in 2020. In 2014, the respective
local maxima (22 ‰ and 10 ‰ for $\delta^{15}N\text{-}NO_3^-$ and $\delta^{18}O\text{-}NO_3^-$) were shifted to km 35. Further downstream, isotope values
decreased, except for a slight increase in the outermost marine samples (Fig. 2d and Fig. 2e).
In 2020, we also measured dissolved nitrous oxide concentration. Measured values ranged between equilibrium concentrations
(~9 nmol L⁻¹) and supersaturation of up to 40 nmol L⁻¹ at km 0, which corresponded to a saturation of 400 %. Nitrous oxide
then decreased downstream to ~ 14 nmol L⁻¹ (140 %) at km 30 and then increased to a local maximum of 21 µmol L⁻¹ (210 %)
in the Tidal River/Dollard Reach transition at stream km 35. Further downstream, nitrous oxide decreased to near equilibrium
concentration towards the North Sea (Fig. 2f).
**3.2 Suspended Particulate Matter properties**
SPM concentration was highest in the Tidal River, reaching values of 2100 mg L⁻¹ in 2014 and 1600 mg L⁻¹ in 2020. SPM
concentration decreased at the beginning of the Dollard Reach region (Fig. 2i). The $\delta^{15}N$-SPM values showed considerable
scatter (Fig. 2l): around 5 ‰ in the Tidal River/Dollard Reach, and 9 ‰ in the Middle Reaches. In the Outer Reaches,
$\delta^{15}N$-SPM dropped again to ~5 ‰. In 2014, $\delta^{15}N$-SPM were elevated (8 ‰), but the database during this cruise is relatively
sparse (Fig. 2l).





In 2020, C/N ratios of SPM (Fig. 2j) were relatively stable in the Tidal River (~11) and Dollard Reach, with a slightly lower
value of 9 in the most upstream sample. In the Middle Reaches, C/N ratios decreased, reaching the lowest value of 6.5 in the
most offshore sample. In 2014, C/N values were 11-15 in the Tidal River, increased to values as high as 20 in the Dollard
Reach and decreased to ~ 11 approaching the North Sea (Fig. 2j).
Particular organic carbon fraction (% POC) was high in the most upstream samples in 2020 (Fig. 2k), decreased to 4.5 % and
remained relatively stable in the Tidal River and Dollard before it increased in the Middle and Outer Reaches up to 11 %. In
2014, the values in Tidal River and Dollard were comparable, but we found a decreasing trend downstream, with a low POC
fraction of ~3 in the outermost sample (Fig. 2k).

**3.3 Dissolved oxygen concentration in the Ems estuary**

In surface water, oxygen concentrations in the Tidal River section were low during both cruises, and increased downstream
with rising salinity. The lowest values were measured in the Tidal River, where the minimum oxygen concentration was
~72 µmol L$^{-1}$ in 2014 and 76 µmol L$^{-1}$ in 2020 (Fig. 2h) corresponding to a saturation of 27 % and 26 % respectively.
Oxygen profiles showed strong vertical gradients with decreasing concentration in deeper water layers. The extent of hypoxia
in the water column depended on the tidal cycle and location, with lowest bottom water oxygen concentration measured at the
most upstream station at stream km 7.2 during low tide in 2020. Detailed profiles can be found in the supplementary material
(S1).
During the continuous near-bottom oxygen measurements, we found anoxic conditions during both of our cruises that lasted
for several hours over a tidal cycle (Fig. 3). Oxygen concentration was generally low at low tide, and elevated at high tide. In
2014, anoxia developed at stream km 11.8 and 18.5, and highest oxygen concentration in bottom water was only 60 µmol L$^{-1}$
(km 24.5) and 70 µmol L$^{-1}$ (km 11.8). At the beginning of August, oxygen concentration at kilometer 11.8 frequently exceeded
measured values at kilometer 24.5.
In 2020, oxygen concentration in bottom water was higher, and anoxia was only found at stream km 11.8. At all other stations,
oxygen concentration remained above 40 µmol L$^{-1}$ even at low tide.

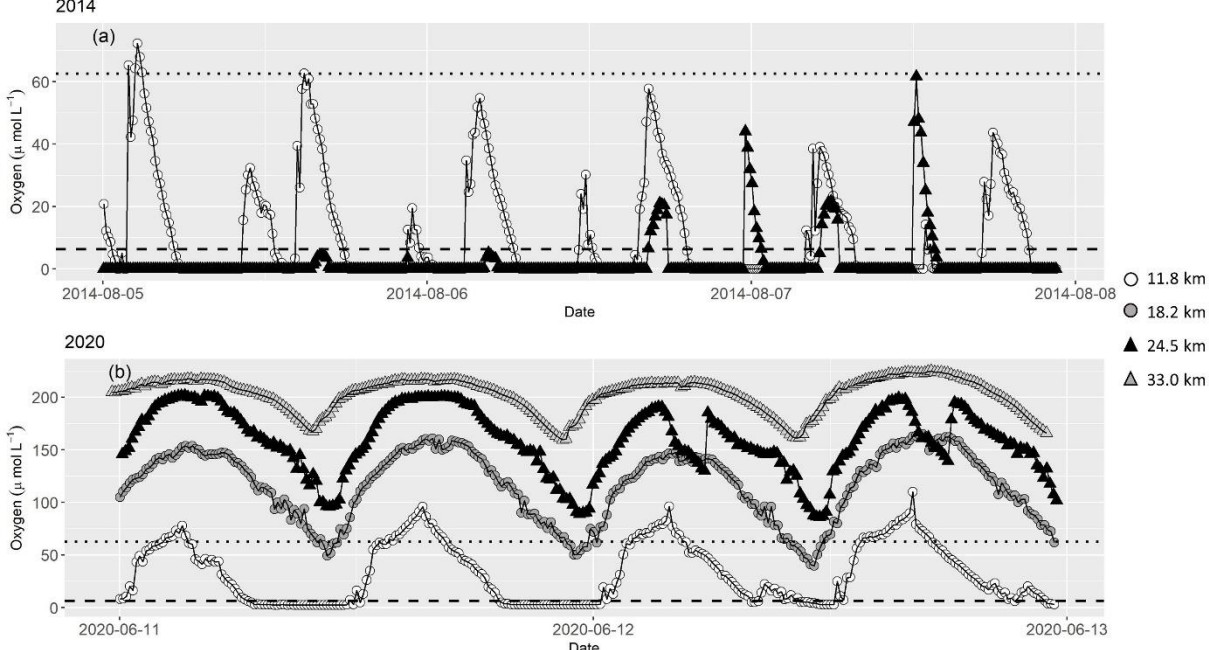

**Figure 3:** *Dissolved oxygen concentration in (µmol L⁻¹) 0.5 m above riverbed during our research cruises in (a) 2014 and (b) 2020 measured continuously at several stations along the Tidal River. Point shapes and colors mark stream kilometer of each sampling station. White points displaying results from a station at stream kilometer 11.8, grey at stream kilometer 18.2, black triangles at stream kilometer 24.5 and grey triangles at stream kilometer 33.0. The dotted line visualizes hypoxic conditions at oxygen concentration of 62.5 µmol L⁻¹ (Diaz et al., 2019). The dashed line shows oxygen concentration (6.25 µmol L⁻¹) under which denitrification occurs (Seitzinger, 1988). Plots (a) and (b) have different y-scales.*

## 3.4 Nitrate mixing

We plotted salinity vs nitrate concentration and nitrate dual isotopes to evaluate mixing properties (Fry, 2002) (Fig. 4). We used the most upstream and downstream samples as end-members for each year. In both years, nitrate concentrations plot below the mixing line in the most upstream region with low salinity in both years, corresponding to an enrichment of $\delta^{15}N\text{-}NO_3^-$ and $\delta^{18}O\text{-}NO_3^-$ in the same region. Above a salinity of 20, a slight nitrate source is present, while isotope values decrease. In 2020, the outermost samples have a slightly enriched isotope signature and nitrate concentration below the mixing line.





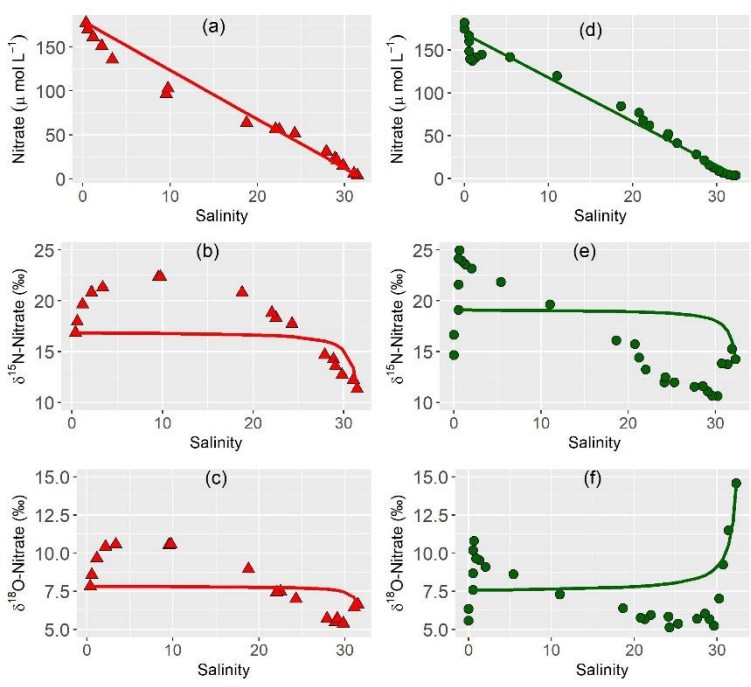

**Figure 4: Nitrate concentrations and isotope values of nitrate plotted versus salinity for (a), (b), (c) in 2014 and (d), (e), (f) in 2020. Lines indicate calculated conservative mixing.**

**3.5 Principal component analysis**

Together, PC1 and PC2 explained about 80 % of total variance in both years. In 2014, PC1 contributed to 66 % and PC2 to 15 %. PC1 and PC2 explained 61 % and 18 % of total variance in 2020, respectively.

Oxygen, pH, C/N ratios, SPM and nitrate concentration contributed largely to PC1 in 2014, just like silicate concentration in 2020 (parameter were not measured in 2014). Temperature, phosphate and nitrite concentration contributed largely to PC2 in both years, so did $\delta^{15}$N-SPM in 2020. Due to few data, $\delta^{15}$N-SPM could not be included into the principle component analysis of 2014. In 2014, PC2 was also heavily influenced by SPM. The PCA overall suggests that the estuary can be divided into three biogeochemically distinct zones, with zone 1 – "Tidal River", zone 2 – " Dollard Reach/Middle Reaches", and zone 3 – "Outer Reaches" (Fig. 5).





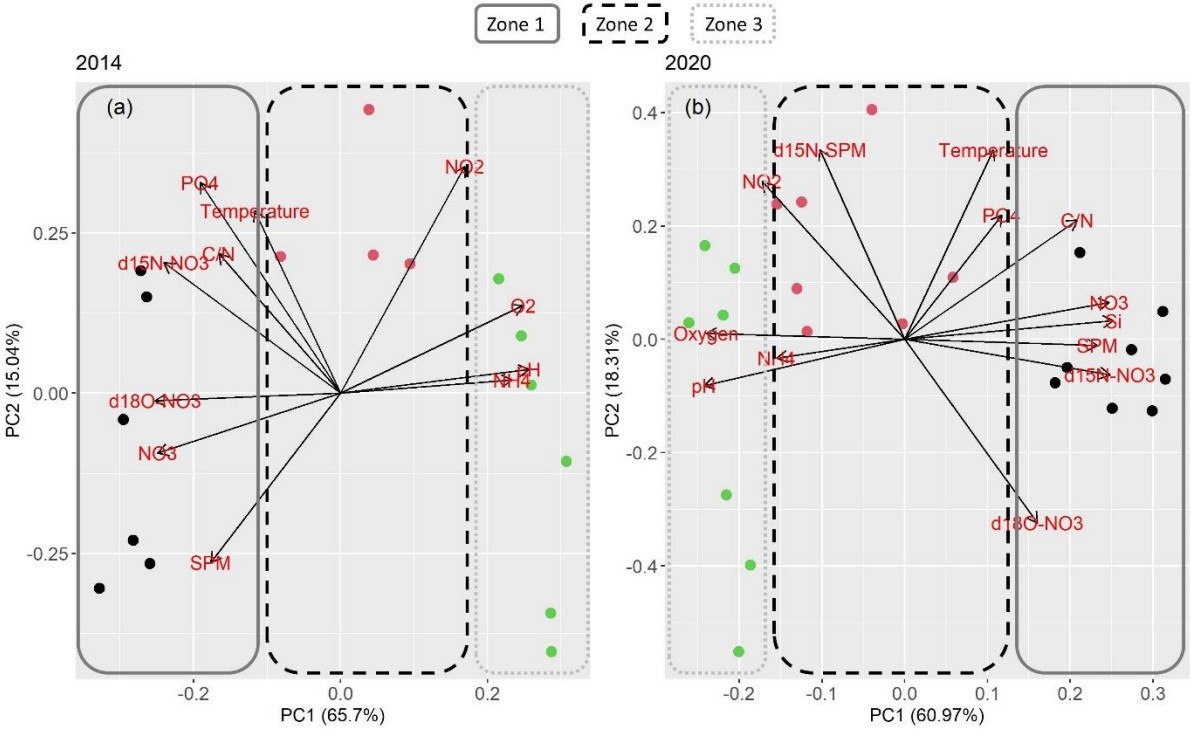

264

**Figure 5: PCA results for (a) 2014 and (b) 2020. Point colors and frames stand for the assignment of the samples into the respective**
**zones. Dark blue points and a straight frame shows samples in zone 1 / Tidal River, red points and a dashed frame samples from**
**zone 2 / Dollard Reach and Middle Reaches. Green points and a dotted frame stand for samples in zone 3 / Outer Reaches.**

## 4 Discussion

### 4.1 Biogeochemical zones in the Ems Estuary

The first goal of this study was to identify distinct zones of nitrogen turnover within the Ems estuary to see if changing
environmental and geomorphological properties affect the occurring processes. The assessment of estuarine mixing curves
showed three zones of different nitrogen turnover along the salinity gradient (Fig. 4).

In both years, 2014 and 2020, nitrate concentration deviated clearly and in a similar manner from the conservative mixing line.
In the upper riverine part of the estuary, nitrate concentration fell below the conservative mixing line, indicating nitrate removal
(zone 1), followed by a zone with nitrate concentration slightly above the mixing line (zone 2) that acted as a net nitrate source.
In the third zone, nitrate mostly followed the conservative mixing line, with nitrate removal and isotopic enrichment near the
marine endmember in 2020, indicating nitrate uptake by phytoplankton.

The PCA support the suggested nitrate zonation taking the other biogeochemical properties into account (Fig. 5). The three
zones were mainly divided according to PC1. Contributing parameters were oxygen, nitrate, C/N, SPM and silicate, which





suggests a tight coupling of nitrate turnover to suspended particulate matter. PC2 helped to differentiate zone 2. Contributing
parameters (temperature, nitrite, and phosphate) suggest a link to nutrient uptake processes.
Based on the location of zones along the Ems (Fig. 1), we see a connection with the geomorphological characteristics of the
Ems estuary. In both years, zone 1 was located in the hyper-turbid Tidal River and the beginning of zone 2 is characterized by
increasing ammonium concentration. In 2014, zone 1 included the Dollard Reach. In 2020, the Dollard Reach was grouped
into zone 2, together with the Middle Reaches. The shift of zone 2 between the cruises may be driven by discharge conditions:
In 2014, discharge was significantly higher than 2020 (about twice the long-term average discharge of 30 to 40 $m^3$ $s^{-1}$ for June
and August) (NLWKN Bst. Aurich and Engels, 2021), which may have led to a shift of zone 2 downstream as also indicated
by the shift in the salinity gradient and SPM concentrations. De Jonge et al. (2014) showed that elevated discharge can relocate
estuarine turbidity maxima downstream. Zone 3 was in the Outer Reaches in 2014 and 2020.
Overall, mixing properties as well as a PCA suggest that there are three distinct biogeochemical zones that act either as sinks
(zone 1 and 3) or sources (zone 2) of nitrate along the Ems. These ones are mainly defined by discharge and suspended
particulate matter (especially PC1).
**4.2 Denitrification in the upper estuary**
Zone 1, the most upstream region acted as a nitrate sink in both years, with nitrate concentrations below the conservative
mixing line and enriched $\delta^{15}N$-$NO_3^-$ and $\delta^{18}O$-$NO_3^-$ values (Fig. 4d and 4e). Potential removal mechanisms are nitrate
respiration or nitrate assimilation.
High SPM values in the hyper-turbid Tidal River and Dollard Reach (Fig. 2i) reduced light availability, limiting primary
production (Bos et al., 2012). Therefore, phytoplankton assimilation in the upper estuary can be ruled out as a relevant nitrate
sink.
Denitrification is a potential nitrate sink that can lead to strong isotope enrichment. Denitrification was a dominant loss pathway
in the 1980s in other temperate estuaries like the Elbe Estuary (Schröder et al., 1995), where sediment denitrification removed
up to 40 % of the summer nitrate load. We found that $\delta^{15}N$-$NO_3^-$ and $\delta^{18}O$-$NO_3$ in the Ems estuary increased with decreasing
nitrate concentration. $\delta^{15}N$-$NO_3^-$ versus $\delta^{18}O$-$NO_3^-$ plot on a slope of 0.5 in both years, which points towards denitrification
(Supplement Material S2) (Böttcher et al., 1990; Mengis et al., 1999; Granger and Wankel, 2016; Wong et al., 2020). A strong
fractionation occurred ($^{15}\varepsilon \sim 24$ ‰, $R^2 = 0.89$ in 2014 and 26 ‰, $R^2 = 0.76$ in 2020). While denitrification in sediments leads
to little to no fractionation due to a diffusion limitation (Brandes and Devol, 1997; Lehmann et al., 2004; Sigman and Fripiat,
2018), water column denitrification has an isotope effect that fits our calculations (Kendall et al., 2007; Sigman and Fripiat,
2018), and can explain the observed patterns.
Water column denitrification occurs under anaerobic to low oxygen conditions in the water column (Tiedje, 1988). According
to Seitzinger (1988), denitrification occurs at oxygen concentration below 6.25 µM. We measured low oxygen concentration
in surface water during both years with lowest concentration of $\sim 70$ µmol $L^{-1}$ (Fig. 2h), which is well above the threshold for
denitrification. However, vertical oxygen concentration profiles and continuous measurements in the estuary in near-bottom





water showed that deeper water became anoxic in both years. Even though these anoxic conditions only developed for a few
hours over a tidal cycle, we conclude that water column denitrification was the responsible nitrate sink mechanism in the Ems
in 2014 and 2020.
Furthermore, denitrification can also occur on suspended particles. Liu et al. (2013) reported the occurrence of denitrification
on suspended particles in oxic waters in a hyper-turbid river. Xia et al. (2016) observed a high oxygen influx around suspended
particles and decreasing oxygen concentration. They suggests that oxygen was consumed by nitrification and/or microbial
respiration close to the particle's surface and thereby provided redox conditions for coupled nitrification-denitrification to take
place. Zhu et al. (2018) detected aggregates of nitrifiers and denitrifiers on SPM in the Hangzhou Bay in China. Similarly,
Sanders and Laanbroek (2018) propose that coupled nitrification-denitrification processes occur in the upper Ems estuary, and
suggested immediate nitrate consumption driven by suspended particles in the water column.
Overall, we find strong evidence for water column denitrification as in the Tidal River / zone 1, likely in the anoxic bottom
waters. Moreover, coupled nitrification-denitrification can add to this nitrate sink in the hyper-turbid Tidal River.
**4.3 Increasing importance of nitrification in the Middle Reaches**
The mixing lines along the estuary displayed a significant shift of nitrogen turnover from the zone 1 (Tidal River) to zone 2
(Dollard Reach/Middle Reaches). Nitrate concentration plotted above the mixing line, indicating a net nitrate source with
lighter nitrate isotope values (Fig. 4).
Nitrate is produced via nitrification, which was no longer oxygen limited in zone 2 due to increasing concentrations compared
to the Tidal River. A positive correlation between nitrite and ammonium, as well as a negative correlation between nitrite and
nitrate for both years indicate nitrate production via nitrification with nitrite as an intermediate product. This is in line with the
findings of Sanders and Laanbroek (2018), who found nitrification in water column and sediments in 2014.
However, there is no clear indication of nitrification in the correlations of nitrate concentration and nitrate isotopes. Nitrate
isotopes were positively correlated with nitrate concentrations, but such a parallel increase usually does not occur during
nitrification. Nitrification produces isotopically depleted nitrate, but the source of $\delta^{15}$N-NO$_3^-$ and $\delta^{18}$O-NO$_3^-$ are in depended
and it increases the overall nitrate pool. At least in 2014, a plot of $\delta^{15}$N-NO$_3^-$ versus $\delta^{18}$O-NO$_3^-$ still plots on a slope of 0.5 in
2014, suggesting that denitrification may still be of importance in this zone. During denitrification, nitrate isotope values and
concentration are also negatively correlated, because denitrification consumes light nitrate and elevates the isotope values in
the remaining pool.
The positive correlation in our study thus is intriguing. It seems likely that denitrification still occurs in parts of zone 2, either
in the oxygen limited conditions in deeper water layers, in the sediments of the adjacent tidal flats (compare to Gao et al.,
2010) or driven by still elevated SPM concentrations of 185 and 230 mg L$^{-1}$ in 2014 and 2020 respectively. The net addition
of nitrate, however, is a clear sign of nitrification.
Accordingly, we aim to explore whether the parallel increase of nitrate concentration and isotope values can be explained by
simultaneous nitrification and denitrification. To identify the influence of both processes, we used a mapping approach inspired



by Lewicka-Szczebak et al. (2017). A detailed description of the open-system mapping approach and figures are shown in the
supplementary material (S2). Briefly, we try to disentangle the influence of nitrification and denitrification in zone 2 based on
the open-system isotope effects, where the slope of the linear relationship between nitrate isotope values and remaining fraction
of nitrate concentration correspondents to the isotope effect (Sigman et al., 2009). The initial values used for the mapping are
derived from the nitrate mixing calculations based on Fry (2002).
For denitrification, we calculated an isotope effect of $^{15}\varepsilon_{DENIT}$ = -26 ‰ in zone 1. For nitrification, the expression of the isotope
effect depends on the abundance of ammonium. As long as ammonium is limiting, we assume that any ammonium is converted
to nitrite and nitrate, so that the apparent isotope effect is that of remineralisation, as long as ammonium concentration is low.
In most parts 2, no ammonium was accumulated. A simultaneous increase of $\delta^{15}$N-SPM, ammonium and nitrite concentration
at stream kilometer 50 in 2020 point towards remineralisation (Fig. 2a, 2b and 2l). Based on $\delta^{15}$N-SPM, we calculated an
isotope effect of $^{15}\varepsilon_{REMIN}$ = -1.2 ‰ ($R^2$ = 0.26), which fits with previous assessments of the isotope effect of ammonification
(Möbius, 2013). We applied this value for nitrification with prior remineralisation. Further downstream, ammonium and nitrite
concentrations increased, so that we assume that remineralisation no longer determines the overall isotope effect of
nitrification. Instead, there was a combined influence of ammonium oxidation with an isotope effect $^{15}\varepsilon$ = -14 to -41 ‰
(Mariotti et al., 1981; Casciotti et al., 2003; Santoro and Casciotti, 2011) and nitrite oxidation with $^{15}\varepsilon$ = +9 to +20 ‰ (Casciotti,
2009; Buchwald and Casciotti, 2010; Jacob et al., 2017). As we measured elevated ammonium and nitrite concentrations, both
processes influenced the fractionation caused by nitrification. Therefore, for total nitrification we assumed a combined isotope
effect of $^{15}\varepsilon_{NITRI}$ = -10 ‰, that we used to describe nitrification in samples with accumulated ammonium and nitrite. This
number is lower than previously measured for ammonium oxidation, and is based on nitrification rate from incubations
performed previously in the Elbe estuary (Sanders, unpublished data).
Based on these input variables, the mapping approach can indeed explain the development of isotope effects and nitrate
concentration. In the most upstream samples, nitrate removal exceeded production: In 2014, denitrification removed
26 µmol L$^{-1}$, and nitrification added 10 µmol L$^{-1}$. In 2020, the mapping approach suggests an addition of 52 µmol L$^{-1}$ and
simultaneous denitrification of 62 µmol L$^{-1}$. In the middle of zone 2 nitrification gained in relative importance with an
approximated production of 10 µmol L$^{-1}$ in 2014 and 20 µmol L$^{-1}$ in 2020, in contrast to denitrification of approximately
3 µmol L$^{-1}$ and 10 µmol L$^{-1}$ respectively. In the most downstream samples, mixing was dominant, and we detected neither
nitrate production nor reduction.
Overall, nitrification and denitrification determined the evolution of nitrate isotopes and concentration in the estuary.
Downstream zone 2, nitrification becomes increasingly important, and the relevance of denitrification ceases. Both processes
lose in importance towards the North Sea, when mixing turns to be the most important process.
**4.4 Mixing and nitrate uptake in the Outer Reaches**
In the Outer Reaches/zone 3 the mixing line shows divergent trends for our two cruises (Fig. 4). While conservative mixing
dominates in 2014, 2020 shows nitrate uptake in the North Sea.





For 2020, a plot of $\delta^{15}N\text{-}NO_3^-$ versus $\delta^{18}O\text{-}NO_3^-$ falls along a slope of 1.5, which points towards simultaneous assimilation and
nitrification (Wankel et al., 2006; Dähnke et al., 2010). The isotope effect $^{15}\varepsilon$ of this drawdown is - 3 ‰, which also is a sign
for assimilation, even though it is at the lower end of values reported for pure cultures (Granger et al., 2004). C/N values close
to Redfield Ratio in 2020 (Fig. 2j) also pointed towards primary production in the Outer Reaches. The stronger signal of nitrate
uptake in June 2020 compared to August 2014 is likely caused by a stronger influence of the spring phytoplankton bloom in
the Outer Reaches (Colijn, 1983; Colijn et al., 1987; Brinkman et al., 2015) fueled by continuous nutrient supply from the
estuary.
In the mixing plot (Fig. 4), the outermost isotope samples of our cruise in 2020 fall on the conservative mixing line. The good
fit is caused by the calculation with a marine endmember that has an isotopically enriched signature in comparison to average
global values (Sigman et al., 2000, 2009) and North Sea winter values of 5 ‰ (Dähnke et al., 2010). The increase of the isotope
signature shows that fractionation takes place, likely due to assimilation.
In contrast to the biogeochemical active inner zones, mixing dominated nitrate distribution in the Outer Reaches of the estuary
in 2014. In 2020 however, the Outer Reaches were a nitrate sink due to ongoing primary production in the coastal North Sea.
**4.5 SPM as driving force of the spatial zonation**
We identified three zones of nitrogen turnover along the estuary, which differ significantly in their coastal filter function. The
Tidal River was a nitrate sink with dominating water column denitrification. In the Middle Reaches, nitrification gained in
importance, turning this section in a net nitrate source. In the Outer Reaches / zone 3, mixing gained in importance but with a
clear nutrient uptake in 2020. Other estuaries with high turbidity show strong denitrification zones as well (Ogilvie et al., 1997;
Middelburg and Nieuwenhuize, 2001). This finding and our analysis of the PCA and dominant nitrogen turnover processes
suggest that the overarching control on biogeochemical nitrogen cycling and zonation may be suspended particulate matter.
Channel deepening led to tidal amplification and an increased sediment transport in the estuary (Winterwerp et al., 2013; De
Jonge et al., 2014; Van Maren et al., 2015b, a). Between 1954 and 2005, SPM concentration increased on average 2- to 3-fold,
and even 10-fold in the Tidal River. The turbidity maximum extended to a length of 30 km and moved upstream, into the
freshwater Tidal River (De Jonge et al., 2014).
High C/N ratios (Fig. 2j), as well as a low and stable particular organic carbon (POC) fraction of the SPM in this region (~
4.5 %) in the Tidal River and Dollard Reach indicate low organic matter quality and a large contribution of mineral associated
organic matter of the present organic matter (Fig. 2k). In 2014, C/N ratios were extremely high, and uncharacteristic for
estuarine environments. We attribute this to a potential influence of peat soils or peat debris in sediments (Broder et al., 2012;
Loisel et al., 2012; Wang et al., 2015; Papenmeier et al., 2013), which may have been washed into the river due to high
discharge. The extremely high C/N ratios should nonetheless be treated with caution, as we cannot entirely rule out sampling
artifacts.





Nonetheless, and regardless of organic matter origin, degradation of organic carbon leads to anoxic conditions in the Tidal
River. Even though the low quality of organic matter fuels only low degradation rates with POC fractions of ~ 3 % (Fig. 2k),
the extremely high POC concentration (> 4000 µmol L$^{-1}$) support the intense oxygen depletion and anoxic conditions in the
Tidal River. This indicates very refractory material. Talke et al. (2009) found oxygen depletion rates proportional to SPM
concentrations in the Ems estuary. Moreover, high SPM concentrations depress  primary production throughout the inner
estuary due to light limitation and leads to a dominance of heterotrophic processes (Bos et al., 2012).
With decreasing SPM concentration, oxygen concentration increases, and the relevance of denitrification ceases in comparison
to nitrification. In zone 2 at the transition between Dollard Reach and Middle Reaches, C/N ratios start to decrease, indicating
the input of fresh organic matter entering the estuary from the North Sea (Van Beusekom and de Jonge, 1997, 1998) fueling
nitrification in zone 2. Although, the quality of the organic matter improves, oxygen depletion decreases due to reduced SPM
concentrations leading to lower POC concentrations in comparison to the Tidal River. Towards the North Sea, low SPM
concentration in the Outer Reaches enable deeper light penetrations supporting local primary production (Liu et al., 2018;
Colijn et al., 1987) as also supported by a slight chlorophyll maximum in the Outer Reaches (S4). Given the ongoing import
of organic matter from the North Sea to the Wadden Sea and adjacent estuaries, this primary produced organic material
probably fuels the remineralisation process in the inner estuary.
Changing discharge conditions can lead to a spatial shift of zones within the estuary. De Jonge et al. (2014) already showed
that elevated discharge relocate ETM in downstream position. As we identified SPM concentration as one of the most important
controls on nitrogen turnover in the Ems estuary, we assume that the zones will move with shifting SPM concentration along
the estuary.
Overall, we find that the interplay of nitrification/denitrification and assimilation is governed by SPM concentration along the
Ems estuary. We expect that changing discharge can lead to spatial offsets in SPM concentrations and thus influence the spatial
segregation nitrogen turnover processes.
**4.6 Nitrous oxide production and its controls in the Ems estuary**
So far, we elucidated nitrogen turnover in the Ems Estuary. We found that nitrification and denitrification vary spatially in
importance. Both processes can produce nitrous oxide, and we accordingly found nitrous oxide peaks in the estuary in areas
with significant differences in their nitrogen turnover. Nitrous oxide was measured only in 2020, thus we will use the high-
resolution data from this cruise to examine the importance nitrification and denitrification for nitrous oxide production along
the estuary. We will also discuss controls that favor the emergence of nitrous oxide production areas.
The calculated average sea-to-air flux of 0.35 g-N$_2$O m$^{-2}$ a$^{-1}$ results in a total nitrous oxide emission of 0.57 × 10$^8$ g-N$_2$O a$^{-1}$
along the Ems estuary. In June 1997,  a significantly higher average sea-to-air flux density of 1.23 g-N$_2$O m$^{-2}$ a$^{-1}$ was measured
(Barnes and Upstill-Goddard, 2011), which amounted to an annual nitrous oxide emission of 2.0 × 10$^8$ g-N$_2$O a$^{-1}$ over the





entire estuary. Upscaling from a single cruise to an entire year is somewhat questionable, but it interesting to note that the
emissions may have halved since the 1990s. Furthermore, our results as well as those from 1997 were obtained from a single
survey in June making the comparison intruding. Since the 1990s, the DIN load of the Ems estuary was significantly reduced
due to management efforts (Bos et al., 2012). Phytoplankton biomass in the Outer Reaches (Station Huibertgat Oost, Van
Beusekom et al., 2018) decreased in response to decreasing nutrient loads, possibly contributing to the observed lower $N_2O$
emissions. However, this hypothesis requires verification in the future.
The nitrous oxide concentrations observed in 2020 can be linked to the prevailing biogeochemical conditions. The first nitrous
oxide maximum was located in the upstream region (stream kilometer 0). In this area, we identified water column
denitrification as the dominant nitrogen turnover process, and we found relatively low pH values and high nitrate concentration.
In their summary paper about nitrous oxide in streams and rivers, Quick et al. (2019) found that these factors are favorable for
nitrous oxide production via denitrification. Intermittent oxygen hypoxia and anoxia in the different water depths also enhance
nitrous oxide production in the Tidal River, which is in line with our tidal oxygen measurements in the Ems. Several studies
also showed a positive correlation between nitrous oxide concentration and SPM concentration (Tiedje, 1988; Liu et al., 2013;
Zhou et al., 2019), and SPM concentration was also highest in this region of the Tidal River. Altogether, we suggest that the
Ems is well suited as a region with extremely high nitrous oxide production, triggered by high nutrient loss, intermittent anoxia,
and high SPM loads.
Further downstream, nitrous oxide concentrations decrease, along with oxygen concentrations, reaching a minimum around
km 22. The simultaneous reduction of nitrous oxide and oxygen concentration at first sight seems counterintuitive, but it may
be caused by complete denitrification that produces $N_2$ instead of nitrous oxide (Knowles, 1982).
Based on our data, we cannot clearly say whether the source of nitrous oxide production was in the water column or in
sediments. Other studies, e.g.  in the muddy Colne estuary found high nitrous oxide production due to denitrification, but
assigned nitrous oxide production only to the sediments (Ogilvie et al., 1997; Robinson et al., 1998; Dong et al., 2002).
Sedimentary denitrification in our study may have contributed to this first nitrous oxide maximum. The beginning of ebb tide
during our campaign may have enhanced outgassing of nitrous oxide from the sediment, and low water levels may have caused
a mechanical release of nitrous oxide from the sediments caused by our research vessel. Thus, zone 1 is an important nitrous
oxide production zone, but the measured nitrous oxide concentration might in parts be affected by sedimentary processes and
might overestimates nitrous oxide production in the water column.
The second nitrous oxide maximum occurred around stream kilometer 35 at the transition between Tidal River and Dollard
Reach. In this area, our mapping approach indicates simultaneous denitrification and nitrification. The nitrous oxide peak
coincides with an increase of ammonium and nitrite concentration as well as a slight rise in nitrate concentration, indicating
the onset of nitrification in the water column.



In contrast to condition leading to the first nitrous oxide peak, not enough fresh organic matter seems to be present in the
transition area to support nitrous oxide production. Lower SPM concentrations with comparable low POC fraction leads to
lower remineralisation rates and higher oxygen levels. Low organic matter availability and increasing oxygen concentration
favor nitrous oxide production via nitrification (Otte et al., 1999; Sutka et al., 2006). Similarly, Quick et al. (2019) summarized
aerobic or oxygen limited conditions with low organic carbon availability favorable for nitrous oxide production via
nitrification. As our data suggests additional denitrification, we speculate that in possible anoxic microsites on suspended
particles and anoxic deeper water layers, denitrification may have contributed to nitrous oxide production. Overall, we assume
that nitrification and denitrification jointly added to nitrous oxide production in this region.
In summary, we find that two nitrous oxide production hotspots exist in the Ems estuary. SPM plays a big role controlling the
nitrous oxide production along the Ems estuary. In the upstream region, where oxygen depletion occurs due to immense SPM
concentration, denitrification produces nitrous oxide. At the transition zone between Tidal River and Dollard Reach, SPM
concentration is lower, leading to higher oxygen concentration and nitrous oxide production via nitrification. Denitrification
prevails in deeper water layers where oxygen concentration is low, and possibly in anoxic microsites close to particles.
**Conclusion**
Overall, the Ems estuary acted as a nitrate sink in both years. However, we found that three distinct biogeochemical zones
exist along the Ems. Stable isotope changes point towards water column denitrification in the turbid water column of the Tidal
River. In the Dollard Reach/Middle Reaches nitrification gains importance turning this section of the estuary into a net nitrate
source. Nitrate uptake occurs in the Outer Reaches due to primary production in the coastal North Sea, in August 2014 mixing
dominated. Our analysis of the dominant nitrogen turnover processes suggest that SPM concentration and the linked oxygen
deficits exert the overarching control on biogeochemical nitrogen cycling, zonation and nitrous oxide production in the Ems
estuary.
Changing biogeochemical conditions can significantly alter estuarine nutrient processing. Deepening of rivers happens not
only in Germany (Kerner, 2007; Schuchardt and Scholle, 2009; De Jonge et al., 2014; Van Maren et al., 2015b) but worldwide
(e.g. Van Maren et al. 2009; Winterwerp et al. 2013; Cox et al. 2019; Grasso and Le Hir 2019; Pareja-Roman et al. 2020), and
this can change SPM loads and composition in estuaries. Increased SPM loads can enhance denitrification, but also trigger
nitrous oxide production and enhance oxygen-depleted zones. Thus, the interplay of SPM with riverine nutrient filter function
and nitrous oxide emissions should be evaluated. The common practices of deepening and dredging affect SPM and this pose
a direct link between pressing social and ecological problems in coastal areas.





**Data availability**

Data will be available under coastMap Geoportal (www.coastmap.org) connecting to PANGAEA. (https://www.pangaea.de/)
with DOI availability.

**Author contribution**

GS, TS and KD designed this study. GS did the sampling, sample measurement and analyses for the cruise of 2020 as well as
the data interpretation and evaluation. TS did the sampling and sample measurement for the cruise in 2014. YV provided the
oxygen data and correction from the FerryBox. AS provided the oxygen data from German Federal Institute of Hydrology.
KD, AS, YS, JB and TS contributed with scientific and editorial recommendations. GS prepared the manuscript with
contributions from all co-authors.

**Competing interest**

The authors declare that they have no conflict of interest.

**Acknowledgment and Funding**

This study was funded by the Deutsche Forschungsgemeinschaft (DFG, German Research Foundation) under Germany's
Excellence Strategy – EXC 2037 'CLICCS - Climate, Climatic Change, and Society' – Project Number: 390683824,
contribution to the Center for Earth System Research and Sustainability (CEN) of Universität Hamburg.
We thank the crew of R/V Ludwig Prandtl for the great support during the cruises. Thanks to Leon Schmidt, who measured
the nutrients and helped during the field work. Phillip Wiese is gratefully acknowledge for the isotope analyses of the 2014
sampling. We are thankful for the FerryBox Team: Martina Gehrung for the preparation and Tanja Pieplow for the oxygen
measurements. Thanks to the working group of Biogeochemistry at the Institute for Geology for measuring C/N ratios, POC
fractions and $\delta^{15}$N-SPM values.

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
