# Peer review of "Suspended Particulate Matter drives the spatial segregation of nitrogen turnover along the hyper-turbid Ems estuary"

_Biogeosciences, 2021_

## Author Response (AR2)

**Author responds**

15

**1 Review comment (RC1) - 04.01.2022**

We thank the reviewer for their constructive and helpful review of our paper. In following, we will reply to the individual comments. Reviewer comments are written in italics, our answers are kept in plain font.

5 I was a bit confused when the methods section talked about FerryBox sampling (Line 92) and equilibrator measurements of nitrous oxide (Line 132), yet the datasets only show discrete samples for the sampling stations along the 100 km transect (Figure 2). To be clear, I like the data presentation in Figure 2 because its clear and easy to follow, but at the moment the Methods section highlights sampling that is not reflected in the results section.

We agree that the current presentation of the FerryBox and nitrous oxide data does not represent the continuous measurements

10 during our campaigns. We changed the figure by plotting data points for 10 min averages along the transect for the continuous measurements (nitrous oxide, oxygen and salinity). We also clarified in the figure caption that we chose to plot 10 min average.

| Lines   | Change                                                                                                  |
|---------|---------------------------------------------------------------------------------------------------------|
| 180     | Changing Fig. 2 with 10 min means for the continuous measurements of nitrous oxide, salinity and oxygen |
| 184-185 | Adding "For clarity, only 10 min means are plotted for the continuous measurements of nitrous oxide     |
|         | (f), oxygen (h) and salinity (g)." to the figure capture                                                |

The only continual measurements are a snapshot of dissolved oxygen concentrations over a 2-3 day period for fixed locations (Figure 3). Also, Figure 3 is less easier to follow than the other figures in the manuscript and I think future readers would appreciate efforts to make it more interpretable.

**We modified the figure caption by adding a brief sentence on sampling points ("In 2014, oxygen concentration was measured at two station at stream kilometers 11.8 and 24.5. In 2020, additional measurements were done at stream kilometers 18.2 and 33.0"). Further, we removed the lines from the figure 3 and added color to the plot to help with the interpretation.**

| Lines   | Change                                                                                                 |
|---------|--------------------------------------------------------------------------------------------------------|
| 241     | Changing Fig. 3 without lines and color                                                                |
| 244-245 | Adding "In 2014, oxygen concentration was measured at two stations at stream kilometers 11.8 and 24.5. |
|         | In 2020, additional measurements were done wat stream kilometers 18.2 and 33.0" to the figure capture  |

- 20 It wasn't clear to me whether the authors interpret the datasets obtained six years apart (in 2014 and 2020) to be sufficiently similar that they can be considered a sampling replicate or whether there are differences between 2014 and 2020 that indicate changes to the N cycling. Looking at Figure 2, there appear to be differences in nitrite concentrations and also the isotopic composition of nitrate. Is this noteworthy to the readers? Discussion of this could be included when discussing differences in the PCA plots on Figure 5?
- 25 Thanks to the referee for the comment. We interpret the data of both sampling campaigns not as sampling replicates, but as sufficiently similar for comparison to evaluate the zonation of the estuary. With a six-year time difference, we cannot ensure identical sampling and measuring conditions for both cruises that would be necessary for true replicates. The PCA analysis showed that nitrogen turnover was comparable in both years. However, there are distinct differences (such as the occurrence of assimilation in the outer estuary, which is much more distinct in 2020) between the cruises that we tried to address in the
- 30 manuscript.

Seasonal and interannual variation may cause differences in dissolved inorganic nitrogen distribution and nitrate stable isotope composition. Water temperature and discharge were significantly higher during our cruise in 2014 than in 2020, reflecting early and late summer. The offset of nitrate isotopes fits relatively well with the observed shift of salinity, oxygen and ammonium concentration. In the manuscript, we argue that these shifts may be driven by the increased discharge (Line 285 -

35 290). The increased nitrite concentration in 2014 may be caused by enhanced temperature that fuels microbial turnover.

In general, we see the same nitrogen turnover processes along Ems estuary during both cruises. There is no evidence hinting significant changes that were not caused by seasonal or single event effects. The PCA independently confirms the same zones of nitrogen turnover for each year. The loadings of the principle components were also similar for both cruises.

In summary, the two cruises allow a more robust assessment of general N-cycling patterns in the Ems than a single campaign 40 would, but they can certainly not be regarded as replicates.

We addressed similarities and differences in the revised manuscript in the chapter "4.1 Biogeochemical zones in the Ems estuary" by adding the following paragraph (L284 - 287).

| Lines   | Change                                                                                                    |
|---------|-----------------------------------------------------------------------------------------------------------|
| 284-287 | Adding "The PCA analysis showed that nitrogen turnover was comparable in both years. However, there       |
|         | are distinct differences between the cruises. Seasonal and interannual variation may cause differences in |
|         | dissolved inorganic nitrogen distribution and nitrate stable isotope composition. The PCA independently   |
|         | confirms comparable zones of nitrogen turnover for both years. The principle components loadings were     |
|         | also similar for both cruises."                                                                           |

**Line 362 The authors justify using an isotope effect of 10 ‰ based on unpublished data. This is should be changed. The 45 authors can always deposit the data in a free public database e.g. zenodo, and cite the doi.**

The data was presented at the EGU in 2014 (Sanders, T.; Daehnke, K.: N-Isotope fractionation of nitrification in the tidal influenced Elbe River estuary, Germany. In: European Geosciences Union General Assembly, EGU 2014. Wien (A), 27.04.-02.05.2014, 2014.). We added the conference abstract as a reference. In addition, we are currently working on a publication of the data as a short communication. Inclusion of the data in a data base is difficult, as these are results from incubation

50 experiments that needs further method explanation, and this format is not easily included in a free public database like, e.g., PANGAEA.

| Lines   | Change                                      |
|---------|---------------------------------------------|
| 376     | Adding reference "Sanders and Dähnke, 2014" |
| 704-706 | Adding reference                            |

Line 443 'Furthermore, our results as well as those from 1997 were obtained from a single survey in June making the comparison intruding' This is just a small language error, the authors should change intruding to intriguing or another would be better reflect their interview.

**55 word to better reflect their intention.**

We corrected this to intriguing in the revised version.

| Lines | Change                               |
|-------|--------------------------------------|
| 351   | Changing "intruding" to "intriguing" |

Figure 1. Is it possible for the authors to either indicate on the map, the four sections that are referred to in the text, or draw a transect below the map that indicates the four zones?

60 We included indications for the four sections in a new version of our map (Fig. 1).

| Lines | Change                                                                                               |
|-------|------------------------------------------------------------------------------------------------------|
| 85    | Changing Fig. 1 with indications of the Tidal River, Dollard Reach, Middle Reaches and Outer Reaches |

**Was the O2 sensor data included in the Supplementary Material? I don't think I saw it there.**

The oxygen data is not part of the supplement materials. We did not include it, because we used data provided by the German Federal Institute of Hydrology (Bundesanstalt für Gewässerkunde – BfG). The BfG itself plans the publication of the data soon

65

in "BfG-Report No. 2077". As soon as the report is published, we will provide links to the data. We hope this is sufficient, as all measured data points ware presented in the figures 3 and S1.

**2 Review comment (RC2) - 10.02.2022**

We thank the reviewer for their constructive and helpful comments and suggestions about our paper. In the following paragraphs, we will reply to the individual comments. The reviewer comments are written in **bold** italics, our answers are kept in the following for the second second

70 in plain font.

This paper presents an interesting and detailed study on the inorganic nitrogen dynamics in a hyperturbid estuary. The study is original as it combines a classical biogeochemical approach based on concentration vs salinity profiles with an indepth use of isotopic data. Overall, the authors use a quite complex method to estimate nitrate transformation amounts (denitrification and nitrification) based on isotope data for different zones. However the authors do not compare these

- 75 values to the overall nitrate concentration in the estuary. In other words, are the observed processes significant or not at the scale of the estuary? And is the estuary, as a whole, a net source or sink for nitrate? If I'm correct, production/consumption quantities estimated by the mapping approach are quite small compared to the NO3 concentration: a net overall consumption of 9  $\mu$ M (-16 + 7  $\mu$ M) in 2014 and a net 0 in 2020 (-10+10  $\mu$ M) for zone 1 and 2 together. When compared to the initial 150-160  $\mu$ M of nitrate, transformations are small or even unsignificant. For this, I could suggest to
- 80 complete the analyses based on isotopes with a more classical nitrate mass balance approach.

First, we want to thank the reviewer for the comment. A mass-balance approach, as suggested, is certainly valuable. Unfortunately, we do not have the data to perform an adequate mass balance calculation, as this would require either areal process rates, or residual flow into the North Sea at the mouth of the estuary. In a tidal estuary, such flow measurements are difficult to obtain, and the effects of changing tides and residence times must be considered, which would require a demanding

85 modelling exercise that alone could represent another publication. We do fully agree that a detailed mass balance is a worthwhile approach, which should be followed up on in the future. Potentially, this can be done by extrapolating areal (or volumetric) process rates, similarly to Deek et al.(2013).

However, our intention was to investigate the effects of biogeochemistry and morphology on individual turnover processes (e.g., denitrification/nitrification). A detailed mass-balance is beyond the scope of our study.

90 We do agree, though, that our data do not allow solid conclusions on the overall source or sink role of the estuary. We removed these statements in the revised version and referred to source or sink functions of individual sections of the estuary. In these sections, we do see significant net deviations in comparison to conservative mixing, which naturally will affect the overall input if their activity ceases.

| Lines | Change                                                                               |
|-------|--------------------------------------------------------------------------------------|
| 21    | Removing: "Overall, the Ems estuary acted as a nitrate sink in both years. However," |
| 498   | Removing: "Overall, the Ems estuary acted as a nitrate sink in both years. However," |

**L13 – precise "their morphology have been changes" in accordance with the type of changes you cite. Other type of change**

**95 – like chemical or biological changes have also occurred.**

We specified that morphology had changed, following to the suggestion of the reviewer.

| Lines | Change                               |
|-------|--------------------------------------|
| 13    | "Their morphology" instead of "They" |

L32-32: Maybe precise you focus on morphological change here. Other types of changes may also arrise such as wastewater treatment in the basin or fertilizer regulation, etc... like you write later.

We specified that we are referring to changes in morphology in this text passage.

| Lines | Change                                                                   |
|-------|--------------------------------------------------------------------------|
| 30    | "The morphology of estuaries has been " instead of "Estuaries have been" |

**100 L56: what do you mean by "properties"?**

**L61: what do you mean by "properties"? be more precise**

**L65: idem**

We refer to biogeochemical properties that we measured during our cruises. This includes the nutrients, oxygen and suspended particulate matter concentrations as well as discharge conditions, water temperature and pH. We specified that we refer to biogeochemical properties in the introduction chapter and also mentioned the nitrate isotope composition, nutrient

105 biogeochemical properties in the introduction chapter and also mentioned the nitrate isotope composition, nutri concentrations and suspended particulate matter concentration in particular, as these are the main parameters we look at.

| Lines | Change                                              |
|-------|-----------------------------------------------------|
| 56    | Adding "biogeochemical properties"                  |
| 62    | Adding "suspended particulate matter concentration" |
| 63    | Adding "biogeochemical zonation"                    |
| 66    | Adding "biogeochemical water properties"            |

115

110 In line with the suggestion of the reviewer, we clarified that both cruises were done in the summer. Further, the months of our sampling campaigns are specified in the chapter "2.2 Sampling".

| Lines | Change                           |
|-------|----------------------------------|
| 58    | Adding "summer research cruises" |

**L59: is it not also highly urbanized? – so high population density?**

The catchment of the Ems River is rather characterized by agriculture und land use area than by high population density. Agricultural land-use is dominant in the catchment (80%), whereas urban land use makes up for 8 % of the catchment (FGG Ems, 2015). We added this information into the chapter "2.1 Study site" (Line 71-73).

| Lines   | Change                                                                                             |
|---------|----------------------------------------------------------------------------------------------------|
| 71-73   | Adding "Agricultural land-use is dominant in the catchment (80 %), and urban land use makes up 8 % |
|         | of the catchment (FGG Ems, 2015) with a population density of ~200 km -2 (UBA, n.d.)."  |
| 607-608 | Adding reference                                                                                   |
| 749-750 | Adding reference                                                                                   |

**L71: Can you mention the population density? This info is always interesting when comparing watersheds with each other.**

We thank the referee for the suggestion to add information about the population density. We added the population density of  $\sim 200 \text{ km}^{-2}$  (UBA, n.d.) in the study site description (chapter 2.1) (Line 72 – 73).

| Lines   | Change                                                                                             |  |
|---------|----------------------------------------------------------------------------------------------------|--|
| 71-73   | Adding "Agricultural land-use is dominant in the catchment (80 %), and urban land use makes up 8 % |  |
|         | of the catchment (FGG Ems, 2015) with a population density of ~200 km -2 (UBA, n.d.)."  |  |
| 607-608 | Adding reference                                                                                   |  |
| 749-750 | Adding reference                                                                                   |  |

**L60: increased SPM: compared to what? Previous years? When did this start?**

120 The suspended particulate matter concentration in the Ems estuary increased compared to values measured before deepening and dredging activities in the estuary. De Jonge et al. (2014) found a 2- to 3-fold increase of suspended particulate matter concentration in the lower reaches of the estuary in the timeframe from 1954 to 2005. In the Tidal River even a 10-fold increase was observed. We specified in our text that it is a comparison over time in the introduction chapter. Further information about the timeframe and magnitude of the increase are referred to in chapter "4.5 SPM as driving force of the spatial zonation".

| Lines | Change               |
|-------|----------------------|
| 61    | Adding "since 1950s" |

**125 Figure 1: Locate the Dollar reach on this map as you mention it often in the text**

We included a modified Figure 1 with indications for all mentioned zones: Tidal River, Dollard Reach, Middle Reaches and Outer Reaches.

| Lines | Change                                                                                               |
|-------|------------------------------------------------------------------------------------------------------|
| 85    | Changing Fig. 1 with indications of the Tidal River, Dollard Reach, Middle Reaches and Outer Reaches |

**Figure 2: it is a bit strange that you connect points for nitrous oxide and not for other variables.**

We agree with the reviewer that the data presentation in figure 2 is not optimal. We changed the figure by plotting data points for 10 min averages along the transect for the continuous measurements (nitrous oxide, oxygen and salinity). We also clarified in the figure caption that we chose to plot 10 min average.

| Lines   | Change                                                                                              |
|---------|-----------------------------------------------------------------------------------------------------|
| 180     | Changing Fig. 2 with 10 min means for the continuous measurements of nitrous oxide, salinity and    |
|         | oxygen                                                                                              |
| 184-185 | Adding "For clarity, only 10 min means are plotted for the continuous measurements of nitrous oxide |
|         | (f), oxygen (h) and salinity (g)." to the figure capture                                            |

About the zones: it took me some time to understand you did not speak about "geographical" zones but more about ..? geochemical? Biogeochemical?

We clarified that we refer to the biogeochemical zonation of the estuary. This zonation does fit relatively well with 135 geomorphological characteristics of the estuary. Therefore, we chose to name the zones according to the geographical zones. To prevent confusion, we referred to the zones in the reviewed manuscript with "Denitrification zone" / zone 1, "Coactive zone" / zone 2 and "Outer zone" / zone 3 instead of "Tidal river" / zone 1, "Dollard Reaches/Middle Reaches" / zone 2 and "Outer Reaches" / zone 3. However, the results from our two cruises also show that the biogeochemical zones can move along the estuary depending on parameters like river discharge. In 2014, the discharge of the estuary was significantly higher than

140 in 2020 or than the long-term average for August. Together with intense transport of suspended particulate matter downstream, this very likely caused the extension of the denitrification zone further into the Dollard Reach.

| Lines | Change                                                                                         |
|-------|------------------------------------------------------------------------------------------------|
| 63    | Adding "biogeochemical zonation"                                                               |
| 266   | Removing "with zone 1- "Tidal River", zone 2 – "Dollard Reach/Middle Reaches", zone 3 – "Outer |
|       | Reaches".                                                                                      |
| 273   | Adding "biogeochemical zones"                                                                  |
| 280   | Adding "Outer zone" / zone 3"                                                                  |
| 288   | Adding "biogeochemical zones"                                                                  |
| 292   | Adding "biogeochemical zones"                                                                  |
| 339   | Removing "Tidal River" / zone 1                                                                |
| 337   | Changing "zone 1 / Tidal River" to "Denitrification zone" / zone 1"                            |
| 338   | Removing "zone 2 (Dollard Reach/Middle Reaches")                                               |
| 365   | Changing "correspondents" to "corresponds"                                                     |
| 362   | Adding "Denitrification zone" / zone 1"                                                        |
| 365   | Adding "zone 2"                                                                                |
| 388   | Changing "Outer Reaches / zone 3" to "Outer zone" / zone 3"                                    |
| 404   | Adding "biogeochemical zones"                                                                  |
| 406   | Changing "Outer Reaches / zone 3" to "Outer zone" / zone 3"                                    |
| 429   | Adding "Coactive zone" / zone 2"                                                               |
| 437   | Adding "biogeochemical zones"                                                                  |
| 477   | Adding "Denitrification zone" / zone 1"                                                        |

L274: zone 1 has not the same extension in 2014 than in 2020. In 2014 the nitrate removal extends to salinity 20, while in 2020 only to salinity 4. Zone 1 has thus not a geographical definition. It is not the "upper riverine part" as mentioned – at least not for 2014... L275: idem for zone 2: in 2014, this zone starts at salinity 20 and extends to the mouth while for 2020 it starts at 5 and extends to 25.

In our text, we refer to biogeochemical zones rather than to geographical zones. We made this clearer from the beginning in the introduction of the manuscript to prevent confusion. Therefore, we also renamed our biogeochemical zones, i.e. "Denitrification zone", "Coactive zone" and "Outer zone". We describe and discuss the shift of spatial extension of the zones in chapter "4.1 Biogeochemical zones in the Ems Estuary" (L295 - 300). The shift of zones may be driven by the increased

150 discharge condition in 2014. Which also affects parameters like salinity and the suspended particulate matter concentration.

| Lines | Change                                                                                         |
|-------|------------------------------------------------------------------------------------------------|
| 63    | Adding "biogeochemical zonation"                                                               |
| 266   | Removing "with zone 1- "Tidal River", zone 2 – "Dollard Reach/Middle Reaches", zone 3 – "Outer |
|       | Reaches".                                                                                      |
| 273   | Adding "biogeochemical zones"                                                                  |
| 280   | Adding "Outer zone" / zone 3"                                                                  |
| 288   | Adding "biogeochemical zones"                                                                  |
| 292   | Adding "biogeochemical zones"                                                                  |
| 339   | Removing "Tidal River" / zone 1                                                                |
| 337   | Changing "zone 1 / Tidal River" to "Denitrification zone" / zone 1"                            |
| 338   | Removing "zone 2 (Dollard Reach/Middle Reaches")                                               |
| 365   | Changing "correspondents" to "corresponds"                                                     |
| 362   | Adding "Denitrification zone" / zone 1"                                                        |
| 365   | Adding "zone 2"                                                                                |
| 388   | Changing "Outer Reaches / zone 3" to "Outer zone" / zone 3"                                    |
| 404   | Adding "biogeochemical zones"                                                                  |
| 406   | Changing "Outer Reaches / zone 3" to "Outer zone" / zone 3"                                    |
| 429   | Adding "Coactive zone" / zone 2"                                                               |
| 437   | Adding "biogeochemical zones"                                                                  |
| 477   | Adding "Denitrification zone" / zone 1"                                                        |

In addition, I'm not sure I can really see the third zone in 2014 from the N data in figure 4. L271: the 3 zones are not clearly appearing from figure 4... L276: I don't see a third zone in 2014... but there is one in 2020

Zone 3, the outer zone, shows divergent trends in 2014 and 2020. In 2020, figure 4 showed nitrate uptake in the Outer Reaches of the Ems estuary, which we assigned to assimilation caused by a stronger influence of the spring phytoplankton bloom in the North Sea. In 2014, the identification of the outer zone is more difficult, as the outermost samples follow the conservative mixing line in figure 4. However, these outermost samples are distinct from the prevailing processes in zone 2, because they do not show signs of nitrate production / nitrification, a characteristic of zone 2.

Therefore, a zone 3 exist in 2014 as well as in 2020, but is mostly determined by mixing, which makes a clear distinction based on figure 4 difficult. Our PCA supports the results the separation of samples into zone 2 and zone 3 as is shown in figure 5. We discuss the processes in zone 3 in chapter "4.4 Mixing and nitrate uptake in the Outer Reaches" in detail. To prevent confusion added a paragraph in chapter "4.1 Biogeochemical zonation in the Ems estuary" (Line 280 - 283) to elaborate the separation of zone 2 and zone 3 in more detail.

| Lines   | Change                                                                                                        |
|---------|---------------------------------------------------------------------------------------------------------------|
| 280-283 | Adding: "In 2014, the identification of the "Outer zone" / zone 3 is more difficult, as the outermost samples |
|         | follow the conservative mixing line in Fig. 4. However, these outermost samples are distinct from the         |
|         | prevailing processes in zone 2, because they do not show signs of nitrate production, a characteristic of     |
|         | zone 2."                                                                                                      |

L368-371: I like very much your mapping approach which allows to estimate the amount of NO3 denitrified and produced by nitrification. However, there are quite a lot of assumptions behind this approach. You could go one step further by comparing these number quantitatively to a nitrate mass balance. Does the net decrease or increase in nitrate corresponds to the calculated balance between nitrification and denitrification?

First, we want to thank the reviewer for the comment. A comparison of our mapping approach to a nitrate mass balance would certainly help to strengthen our obtained results. Unfortunately, as we mentioned in a comment above, our data does not allow an adequate mass balance for the tidal Ems estuary. Overall, our mapping approach allows us to estimate the relative

170 an adequate mass balance for the tidal Ems estuary. Overall, our mapping approach allows us to estimate the relative importance of both processes in individual sections of the estuary. To evaluate the results a quantitatively comparison with a nitrate mass balance would be helpful, but is not possible with our data.

To test the validity of our mapping approach, we calculated the net removal based on the integrals below the conservative mixing line. This somewhat more conservative approach, which does not include isotope values, shows that the net mass

175 calculation results of the mapping approach are valid – notably, they are founded on the same data base as the integrals.

The calculation of integrals naturally does not enable us to disentangle the individual contribution of nitrification and denitrification, which is based on isotope effect calculations. We do not see a way to overcome this, and will emphasize in a revised version that the partitioning between these two processes depends on the underlying assumptions for isotope effects.

|                                    | Denitrification    | Nitrification      | Net Production     |
|------------------------------------|--------------------|--------------------|--------------------|
| 2014                               | $(\mu mol L^{-1})$ | $(\mu mol L^{-1})$ | $(\mu mol L^{-1})$ |
| Mapping Approach                   | - 38.4             | + 80.9             | + 42.5             |
| Integrals of nitrate concentration | -                  | -                  | + 46.5             |

|                      | Denitrification    | Nitrification           | Net Production          |
|----------------------|--------------------|-------------------------|-------------------------|
| 2020                 | $(\mu mol L^{-1})$ | (µmol L -1 ) | (µmol L -1 ) |
| Mapping Approach     | - 205.2            | + 242.6                 | + 37.4                  |
| Integrals of nitrate |                    |                         | + 38.2                  |
| integrals of infrate | -                  | -                       | + 36.2                  |
| concentration        |                    |                         |                         |
|                      |                    |                         |                         |

**L394: a small sink (-16 compared to 160 µmol/l ?)**

As we removed statements about the overall role of the estuary, we only refer to the source and sink functions in the individual sections of the estuary.

| Lines   | Change                                                                                                       |
|---------|--------------------------------------------------------------------------------------------------------------|
| 405-407 | "The Tidal River was a nitrate sink with dominating water column denitrification. In the Middle Reaches,     |
|         | nitrification gained in importance, turning this section in a net nitrate source. In the "Outer Zone" / zone |
|         | 3, mixing gained in importance but with a clear nutrient uptake in 2020."                                    |

**L396: remove "as well" or remove "strong". You do not have a strong denitrification zone...**

185 We removed the word "strong" from the sentence.

| Lines   | Change            |
|---------|-------------------|
| 407-408 | Removing "strong" |

L486: "Overall, the Ems estuary acted as a nitrate sink in both years." You do not really show that the Ems acts as a NO3 sink. To do this you should have done a NO3 mass balance for the whole estuary. Hence you show that there is a zone in The Ems where you have a nitrate sink.

We agree with the reviewer. We removed this statement and refer to the activity in the individual zones (see comments above).

| Lines | Change                                                                               |
|-------|--------------------------------------------------------------------------------------|
| 21    | Removing: "Overall, the Ems estuary acted as a nitrate sink in both years. However," |
| 498   | Removing: "Overall, the Ems estuary acted as a nitrate sink in both years. However," |

**Editing comments**

| Lines | Change                                                |
|-------|-------------------------------------------------------|
| 178   | Change "regard" to "look"                             |
| 251   | Change "salinity vs nitrate" to "nitrate vs salinity" |
| 358   | Change "S2" to "S3"                                   |
| 360   | Change "correspondents" to "corresponds"              |
| 453   | Adding "is"                                           |

We thank the reviewer for the thorough revision. We incorporated all suggested editorial changes.

**3 References**

Deek, A., Dähnke, J., van Beusekom, J. E. E., Meyer, S., Voss, M., and Emeis, K.-C.: N2 fluxes in sediments of the Elbe Estuary and adjacent coastal zones, Mar. Ecol. Prog. Ser., 493, 9–21, https://doi.org/10.3354/meps10514, 2013.

FGG Ems: Hochwasserriskiomanagmentplan 2015 - 2021 für den deutschen Anteil der Flussgebietseinheit Ems gemäß §75 WHG, Flussgebietsgemeinschaft Ems, 2015.

Sanders, T.; Dähnke, K.: N-Isotope fractionation of nitrification in the tidal influenced Elbe River estuary, Germany. In: European Geosciences Union General Assembly, EGU 2014. Wien (A), 27.04.-02.05.2014, 2014.

200 UBA, G. central environmental authority: Nationaler Teil der internationalen Flussgebietseinheit Ems, https://www.umweltbundesamt.de/sites/default/files/medien/2466/dokumente/steckbrief\_ems.pdf.

205